# Vaccine Strategies for Human Papillomavirus-Associated Head and Neck Cancers

**DOI:** 10.3390/cancers14010033

**Published:** 2021-12-22

**Authors:** Jade Z. Zhou, Jessica Jou, Ezra Cohen

**Affiliations:** 1Division of Hematology and Oncology, School of Medicine, University of California, San Diego, CA 92037, USA; 2Division of Gynecologic Oncology, Moores Cancer Center, University of California, San Diego, CA 92037, USA; j1jou@health.ucsd.edu; 3Division of Hematology and Oncology, Moores Cancer Center, University of California, San Diego, CA 92037, USA; ecohen@health.ucsd.edu

**Keywords:** vaccines, human papillomavirus, oropharyngeal cancer, head and neck cancers

## Abstract

**Simple Summary:**

Human papillomavirus (HPV) is recognized as a significant risk factor for head and neck cancers worldwide, and it is the most common cause of oropharyngeal cancers in the United States. Here, we review the incidence and pathogenesis of HPV-related cancers, the development and approval of HPV prophylactic vaccines, and the use and effectiveness of HPV vaccines around the world. Furthermore, we discuss advances in the development of HPV therapeutic vaccines as well as its associated challenges.

**Abstract:**

The rising incidence of oropharyngeal squamous cell cancers (OPSCC) in the United States is largely attributed to HPV. Prophylactic HPV vaccines have demonstrated effectiveness against oral infection of HPV 16 and HPV 18. We review the global epidemiology and biology of HPV-related cancers as well as the development of HPV vaccines and their use worldwide. We also review the various strategies and challenges in development of therapeutic HPV vaccines.

## 1. Introduction

Human papillomavirus (HPV) is the most common sexually transmitted disease in the United States and is responsible for an increased incidence of oropharyngeal squamous cell cancers (OPSCC). It is widely accepted that this subset of HPV-associated head and neck cancers differ considerably from those that are not HPV related, with major differences in pathogenesis, epidemiology, and prognosis. The p16 protein is frequently overexpressed in HPV-related OPSCC, and it often serves as a surrogate marker of HPV positivity through detection via immunohistochemistry. HPV-related OPSCC patients are frequently younger and have a distinctly improved response to treatment compared to those unrelated to HPV. Moreover, prophylactic HPV vaccines have been shown to have a strong protective effect against oral infection of HPV 16 and HPV 18, which are contributors to the vast majority of HPV-positive head and neck cancer cases worldwide. Here, we summarize data on the epidemiology of HPV-positive head and neck cancers, the development and approval of HPV vaccines, and their use around the world.

## 2. Epidemiology and Biology of HPV-Related Cancers

Globally, HPV-related malignancies include cervical cancer and other anogenital cancers as well as head and neck cancers, with a large fraction dominated by cervical cancer, which represent 83% of the total burden of cancer attributable to HPV [1]. Head and neck cancers account for nearly 700,000 new cases worldwide annually and are a biologically heterogeneous group of cancers, encompassing cancers originating from the mucosa of the oral cavity, oropharynx, nasopharynx, hypopharynx and larynx [2]. Historically, tobacco and alcohol use were associated with most head and neck cancers. However, decreasing tobacco consumption in the United States in recent decades has led to an overall decline in the incidence of HPV-negative head and neck cancers. Conversely, there has been a shift in the epidemiology of head and neck cancers, with a rise in HPV-mediated oropharyngeal cancer incidence in North America and worldwide [3]. There is strong evidence of a causal link between HPV infection and OPSCC, as the oropharynx is distinctively susceptible to HPV persistence [4,5]. Approximately 70% of patients with oropharyngeal cancer are HPV positive in the United States, with other head and neck cancer sites showing lower HPV prevalence. Analyses of demographics and risk profiles demonstrate a clear contrast between HPV-related and HPV-unrelated OPSCCs. HPV-mediated OPSCC patients are more likely to be significantly younger (<60 years old), male, and of white race [6]. A study by Gillison and colleagues examining lifetime risk factor exposures for HPV-positive versus HPV-negative HNSCC patients demonstrated that HPV-positive HNSCC patients have significantly increased number of lifetime oral sex partners as well as decreased tobacco or alcohol use, greater marijuana use, and improved dentition compared to that of HPV-negative patients [7]. These risk exposures also had a strong cumulative effect. Furthermore, tumor HPV status has been shown to be a strong prognostic factor for superior survival among OPSCC patients, whereas tobacco smoking significantly increases the risk of death [8,9].

HPVs are a family of double-stranded DNA viruses containing proteins involved in viral genome replication (E1, E2 and E4) and assembly (L1 and L2) as well as accessory proteins (E5, E6, and E7) [10]. E6 and E7 are thought to be drivers of carcinogenesis through degradation of p53 and Rb proteins, respectively [11]. The L1 and L2 proteins are structural and form capsids around the viral DNA. There are over 200 types of HPV which are classified as low or high risk according to oncogenicity. HPV types are determined by the degree of homology within the L1 (major capsid protein) gene, and distinct types are defined by differences by more than 10% of the L1 DNA sequence. HPV 16 and HPV 18 are associated with a large majority of HPV-related cancers. HPV 16 is recognized as the most carcinogenic type, accounting for 50% of all cervical cancers, the majority of HPV-related anogenital cancers, and greater than 90% of all HPV-positive head and neck squamous cell carcinomas [10,12].

At least 80% of sexually active young adults acquire HPV at some point during their lifetime, although most people typically clear these infections spontaneously within two years [13]. Among those who do not clear their infections, chronic infection can lead to pre-malignant and malignant lesions in numerous anatomic sites, including the oropharynx [14]. It is unclear why infections persistent in a small subset of the population which later progress to HPV-associated OPSCC. Interestingly, oral HPV 16 DNA was not detectable in most partners of patients with HPV-associated OPSCC, suggesting that most partners clear active infections that they may have acquired [15]. There is an estimated period between HPV infection and OPSCC cancer development of 10 to 30 years by comparing the age of peak prevalence for oral oncogenic HPV infection to median age at OPSCC diagnosis [16].

HPV infects the epithelium of skin and mucous membranes of the anogenital region, involving cervical or anal transformation zones where the stratified and columnar epithelia meet [17]. Studies have shown that certain sexual behaviors such as unprotected oral sex and practicing oro-anal sex play a role in HPV transmission and infection that may result in HPV-associated OPSCC [18,19]. It is thought that susceptibility of this region to HPV-induced transformation could be due to misregulation of viral gene expression. HPV-mediated head and neck cancers mainly arise in the base of tongue and palatine tonsils. The tonsillar crypt cells of the oropharynx, similar to single-layered epithelial cells at the cervical squamocolumnar junction, are more susceptible to carcinogenic transformation [20]. This is thought to be due to strong expression of programmed cell death ligand 1 (PD-L1), which acts to suppress T-cell responses to HPV, thus favoring persistent HPV infection and allowing tumorigenesis. Currently, there is no validated screening test for early detection of HPV OPSCC. Therefore, prevention of high-risk HPV infection is critical in both sexes. 

## 3. Development and Approval of HPV Prophylactic Vaccination

The discovery of virus-like particles (VLPs) from recombinant expression of the major papillomavirus capsid protein L1 in the early 1990s paved the way for vaccine development. VLPs, which can be produced in bacteria, yeast, or insect cells, do not contain an oncogenic viral genome and are immunologically similar to native virions, and animal models consistently showed that vaccination with L1 VLP elicited titers of neutralizing antibodies that protect against viral challenge [21]. Thus far, three HPV vaccines have been introduced against up to nine HPV types and have shown promising results in protecting against HPV infection and related diseases such as genital warts and cancer. The prophylactic effect specifically for oropharyngeal cancers is assumed based on clinical evidence of its prevention of oral HPV infection and HPV-associated cellular changes, including precancerous and benign lesions [22].

As of 2009, both Cervarix, a bivalent HPV vaccine, and Gardasil, a quadrivalent HPV vaccine, have been approved and commercially available. Both are composed primarily of VLPs and were shown to induce sustained serum neutralizing antibody titers of several fold higher than that seen in natural infection [23]. Cervarix contains L1 VLPs against the most common oncogenic types, HPV 16 and 18, with an adjuvant system 04 (AS04) comprising aluminum hydroxide and monophosphoryl lipid A (MPL). As a Toll-like receptor 4 (TLR4) agonist, MPL induces high levels of antibodies [24]. Gardasil, in addition to HPV 16 and 18, also contains VLPs targeting HPV 6 and 11, which cause approximately 90% of genital warts, with an aluminum hydroxyphosphate sulfate adjuvant [25]. 

The phase III efficacy trials of the VLP vaccines demonstrated efficacy in preventing cervical vaccine-related HPV infection and the preneoplastic lesions caused by HPV infections [26]. Other variables include the doses received, baseline cytology, presence of baseline oncogenic HPV types, and whether new infections were counted at the first day of the study or after the third vaccination [27]. All of the trials were randomized, blinded and placebo-controlled trials of young women (mean age 20, range 15–26) of relatively large size (5500–18,500 vaccine recipients). Two phase III studies, FUTURE I [28] and FUTURE II [29], evaluated Gardasil, and the PATRICIA trial [30] and the Costa Rica HPV Vaccine Trial [31] evaluated Cervarix. Both HPV VLP vaccines were established to be safe, highly immunogenic, and induced high peak titers of antibodies with persistent measurable serum antibody responses for years. Among women who were positive for specific HPV types at the time of vaccination, none of the vaccines induced clearance of the infection or disease, demonstrating that these are strictly prophylactic vaccines. The impact of the HPV vaccine on oral disease is limited to demonstrations of a decrease in oral HPV infection after vaccination. A four-year follow up of the Costa Rica HPV Vaccine Trial demonstrated that the bivalent Cervarix vaccine had 93% efficacy in the prevention of oral HPV infection [32]. 

Gardasil 9, a nine-valent vaccine which protects against HPV 6, 11, 16, 18, 31, 33, 45, 52, and 58, was approved by the FDA in 2014, was found to be non-inferior to the quadrivalent HPV vaccine and offers protection against HPV-related cervical, vaginal and vulvar cancers [33]. Gardasil 9 did not prevent infections and diseases related to HPV types beyond the nine types covered by the vaccine. As of 2020, the FDA added oropharyngeal and other head and neck cancers to the list of indications for the Gardasil 9 HPV vaccine based on effectiveness in preventing HPV-related anogenital disease. Table 1 provides an overview of the available HPV prophylactic vaccines, indications, vaccination schedules and FDA approval.

## 4. Use of HPV Prophylactic Vaccination around the World

Studies demonstrate that population effectiveness is highest in those who are vaccinated prior to first sexual contact [34]. A study by Valasoulis and colleagues investigated whether HPV prophylactic vaccination alters HPV-related biomarker expression in women with established minor cervical dysplasia [35]. They found that vaccination of patients with low-grade cytologic abnormalities led to earlier clearance of HPV 16 and HPV 18 DNA-positive infections in comparison to patients who did not receive vaccination. As such, earlier administration of the HPV vaccine appears to be associated with more effective results. A meta-analysis demonstrated the population level impact of female vaccination [36]. After 5–8 years of vaccination, the prevalence of HPV 16 and 18 among girls 13–19 years of age decreased by 83% and decreased by 66% among those aged 20–24 years. A study of over 2600 men and women aged 18–33 years found that in patients who received at least one dose of an HPV vaccine, the prevalence of oral infection with four HPV types was 88% lower than those not vaccinated [37]. 

HPV types in cancer and vaccine efficacy vary geographically. HPV 16 is the predominant type in squamous cell carcinoma, followed by HPV 18. Rarer types vary in their distribution. In most regions, HPV 45, 31 and 33 are the third, fourth and fifth most common genotypes except in Asia, where HPV 58 and 52 are the third and fourth most common types [38]. The efficacy of Gardasil 9 against HPV types associated with cervical cancer was demonstrated to be approximately 88% in Asia, 92% in Africa and North America, 91% in Europe, 90% in Latin America and the Caribbean, and 87% in Australia [39]. Global access remains a true challenge for HPV vaccination. There are several potential societal barriers that influence vaccine coverage and efficacy, such as cost, geographical location, health care infrastructure, education and cultural acceptance [21]. Several high-income countries have introduced the vaccines successfully; however, the vaccination of young people at the global scale remains quite low. Approximately 15% of the targeted global female population have received the full vaccination series, and 20% have received at least one dose [40]. Lower socioeconomic background and lower education are associated with lower vaccine uptake [41]. In the United States, ethnic minorities were found to be more likely to initiate HPV vaccination but less likely to follow through with the full HPV vaccine series, particularly among Black and Hispanic patients [42]. Further educational interventions are needed to increase the acceptance of HPV vaccination and help improve HPV vaccine coverage.

## 5. Therapeutic HPV Vaccine

While preventative vaccines, such as the aforementioned Cervarix and Gardasil 9, have proven to be highly successful in preventing genitourinary and oropharyngeal HPV infections, there are currently no therapeutic options for patients with persistent or established HPV infections. When the HPV virus integrates into the host genome, this often results in loss of some viral genes including several early (E2, E4 and E5) and late (L1 and L2) genes. As a result, L1- or L2-specific neutralizing antibodies generated by prophylactic vaccines are no longer effective against these HPV-infected cells [43].

The E6 and E7 proteins are ideal candidates for vaccine targets for several reasons. First, they are constitutively expressed and obligate oncogene drivers of HPV-associated cancers. Second, the E6 and E7 proteins are also critical for HPV virus induction and continued integration, and thus unlikely to escape immune surveillance through antigen loss. Third, E6 and E7 are foreign proteins and thus would not likely face immune tolerance with vaccination [44]. To eliminate established infections, therapeutic vaccines would need to generate T-cell-mediated immunity by specifically targeting these HPV early antigens. Current vaccines include live-vector, peptide- and protein-, nucleic acid- and whole cell-based vaccines (Table 2).

### 5.1. Live Vector-Based Vaccines

Live vector-based vaccines use genetically attenuated bacterial or viral vectors that carry recombinant DNA encoding the antigen of interest into the host to elicit an immune response. This vaccine vector mimics a natural infection, are highly immunogenic, and takes advantage of the microorganism’s natural ability to infect and incorporate into a host’s genome. They are able to induce a wide range of immune responses including localized reaction and/or systemic humoral cell-mediated immunity. In pre-clinical animal models, Listeria monocytogenes (Lm)-based HPV vaccines have been found to activate both innate and adaptive immune systems [45]. Axalimogene filolisbac (AXAL or ADXS11-001) is a novel therapeutic vaccine that uses an attenuated strain of Lm fused to the non-hemolytic fragment of listeriolysin O and secretes the Lm-LLO-HPV E7 fusion protein targeting HPV-positive tumors [46]. This drug has been evaluated in several clinical trials with various HPV-associated tumors, including two phase I/II trials in patients with head and neck cancers. A phase I/II trial is currently ongoing for patients with recurrent, HPV-positive squamous cell carcinoma of the head and neck randomized to receive AXAL, durvalumab, or both [47] (NCT02291055). Another clinical study of AXAL in previously untreated, surgically resectable, stage II–IV oropharyngeal cancer found increased interferon gamma (IFN-g), tumor necrosis factor alpha (TNF-a), and tumor-infiltrating T cells after receiving the vaccine before and after surgery [48] (NCT02002182). Though these trials have shown promising results, alarming safety signals have also been reported with two patients suffering from systemic listeriosis. Vaccine manufacturers have since withdrawn support for the REALISTIC trial, a phase 1 dose finding trial of Listeria-based vaccine for patients with HPV 16 oropharyngeal carcinoma (NCT01598792).

### 5.2. Peptide/Protein-Based Vaccines

Peptide-based therapeutic vaccines are usually made of synthetic B- or T-cell epitopes that are recognized by their corresponding immune cells, complexed with major histocompatibility complex (MHC) I or II molecules on the surface of antigen presenting cells (APCs). This then activates CD8+ killer T cells and CD4+ helper T cells to interact with B cells to produce specific antibodies against the pathogen [49]. Peptide-based vaccines have the advantage of stability, safety and feasibility of large-scale production. However, these vaccines are in turn, poorly immunogenic and require adjuvantation. One category of peptide-based vaccines contains short peptides (<15 amino acids) that do not require additional processing by professional antigen presenting cells (APCs) and can therefore bind to MHC class 1 molecules of all nucleated cells. However, presentation without co-stimulation can lead to immune tolerance and enhanced tumor growth [50]. Therefore, synthetic long peptides (SLPs) have been developed to harbor both CD4 and CD8 T-cell epitopes but require processing and presentation by professional APCs. A number of synthetic peptide vaccines such as PepCan (NCT02481414) and ISA101 covering the HPV 16 E6 and E7 proteins, respectively, have shown regressions in cervical and vulvar high-grade intraepithelial lesions [51]. These proof of concept studies paved the way to combining immune checkpoint blockage with the tumoricidal vaccine ISA101 in head and neck cancers. A recent single-arm, phase II trial demonstrated an overall response rate of 33% in patients with incurable HPV 16-positive oropharyngeal cancers with the combination of nivolumab and ISA101 [52]. The combination of cemiplimab with or without ISA101b (NCT03669718) and the combination of ISA101b, with a 41BB checkpoint inhibitor, utomilumab (NCT03258008), are now being investigated in a phase II trial in patients with metastatic or recurrent HPV 16-positive oropharyngeal cancer. A phase I/II trials is also underway for use of DPX-E7, a peptide vaccine, in incurable head and neck cancers (NCT02865135). Pre-clinical advances in vaccine adjuvants such as recombinant lipoproteins with Toll-like receptor 2 agonists [53], non-coding, long-chain RNA molecules [54], and a combination of poly IC and costimulatory anti-CD40 antibodies [55] have helped to improve immunogenicity against HPV-associated tumors. 

### 5.3. Nucleic Acid-Based Vaccines

Nucleic acid-based vaccines involve delivering plasmid DNA or mRNA encoding a protein of interest into the host genome. Upon transfection into these cells, the gene of interest can then be expressed and the protein produced after gaining access to the cellular processing machinery. DNA vaccines typically consist of a bacterial plasmid containing a viral promoter, the gene of interest and a termination sequence. After injection, the DNA is taken up by muscle or skin cells. Since myocytes are not professional APCs, the elicited immune response is often weaker and less sustained. When injected intradermally, dendritic cells are professional APCs which lead to MHC class I-associated CD8+ T-cell activation. MHC class II-associated CD4+ helper T cells may also be activated if professional APCs phagocytose transfected somatic cells [56]. This is because MHC class I molecules present products of proteolysis to CD8+ T cells, while MHC class II molecules present products of lysosome degradation to stimulate CD4+ T cells. Macroautophagy therefore delivers intracellular proteins to lysosomal degradation and contributes in this way to the pool of MHC class II presented proteins. DNA vaccines are relatively low cost, shelf stable, elicit both cellular and humoral immunity and have excellent safety and tolerability in human studies. However, DNA vaccines have also been disappointing in their immunogenicity. Cellular uptake of DNA from its surroundings and access to professional APCs is inefficient, usually requiring fusion proteins and carrier molecules. Expression of the encoded antigen DNA sequence also requires codon optimization to enhance transcription and translation of HPV DNA inside nucleoli of a cell [57]. RNA-based vaccines may bear a slight advantage over DNA-based vaccines in this way, as mRNA can be translated directly in the cytoplasm of a cell. Though RNA vaccines are notoriously unstable and may require nanoparticles that protect the mRNA from degradation [58]. The potential concern for integration of DNA in the host cellular genome has never been proven [59]. 

The phase 1b/II safety, tolerability and immunogenicity trial of immunotherapy with MEDI0457 (formerly INO-3112), which is a vaccine combination of synthetic DNA plasmids targeting HPV 16 and 18 E6/E7 antigens (VGX-3100) and a recombinant IL-12 encoding a molecular adjuvant, delivered through electroporation with the CELLECTRA device, found durable peripheral and tumor immune responses [60]. The HARE-40 is a phase I/II trial of a mRNA-based vaccine against the E6 and E7 oncoproteins in combination with anti-CD40 antibody in patients with HPV 16+ head and neck cancers (NCT03418480) that is now active. 

### 5.4. Whole Cell-Based Vaccines

Advances in bioinformatics and genomics have shown increased efficacy in targeting neoantigens via vaccination. Autologous tumor cell-based vaccines ensure that tumor-specific antigens are preserved and used to elicit an immune response. However, in order to target each unique mutanome of every tumor, personalized vaccines are necessary yet expensive and time consuming. Phase I/II trials for personalized vaccines in patients with head and neck cancers include Allovax and MVX-ONCO-1 are currently underway (NCT 02999646, NCT 01998542, NCT02624999). MVX-ONCO-1 uses irradiated autologous tumor cells expressing GM-CSF combined with encapsulated cellular technology that allows continuous production of GM-CSF [61].

**Table 2 cancers-14-00033-t002:** Therapeutic HPV vaccines under development in trials for oropharyngeal/head and neck cancers.

Vaccine	Design (Advantages and Disadvantages)	Clinical Trial and Intervention (Key Results, Where Available)	Publication/Status
Live vector based	Uses genetically attenuated bacterial or viral vectors that carry recombinant DNA encoding the antigen of interest into the host to elicit an immune response. Advantage: highly immunogenic Disadvantage: may cause systemic infection		
AXAL or ADXS11-001	Uses attenuated strain of listeria bacteria fused to the non-hemolytic fragment of listeriolysis O and secretes the Lm-LLO-HPV E7 fusion protein.	Phase I/II in recurrent, HPV-positive squamous cell carcinoma of the head and neck. Randomized to AXAL+ durvalumab (MEDI4736) vs. durvalumab (MEDI4736) alone.	NCT02291055 (active) [45]
		Phase II trial in previously untreated, surgically resectable stage II–IV oropharyngeal cancer. Vaccine is given prior to transoral surgery to investigate T-cell response rate.	NCT02002182 (active) [46]
Peptide-protein based	Made of synthetic immune cell epitopes that elicit immune response from B and T cells through complexing with MHC I and II molecules on antigen presenting cells. Advantage: stable, safe, large-scale production Disadvantage: poorly immunogenic, requires vaccine adjuvant		
ISA101b	Consists of 12 synthetic long peptides derived from the E6 and E7 proteins of the HPV 16 virus.	Phase II trial in patients with metastatic or recurrent HPV 16-positive oropharyngeal cancer received ISA101b with nivolumab. Results: ORR 33%, median PFS 2.7 months, median OS 17.5 months.	NCT02426892 [50]
		Phase II trial in patients with metastatic or recurrent HPV 16-positive oropharyngeal cancer randomized to utomilumab (checkpoint inhibitor) vs. utomilumab with ISA101b.	NCT03258008 (active)
DPX-E7	Consists of synthetic peptide of amino acids 11–19 of the HPV 16 oncoprotein E7.	Phase Ib/II in patients with incurable head and neck cancers to receive DPX-E7 vaccine in a prime-boost schedule.	NCT02865135 (active)
Nucleic acid based	Uses a bacterial plasmid to deliver a segment of DNA or mRNA encoding a protein that targets the E6 and E7 proteins into the host genome. Advantage: low cost, shelf stable, elicits both cellular and humoral immunity, excellent safety and tolerability Disadvantage: poorly immunogenic		
MEDI0457 (INO-3112)	Combination of synthetic DNA plasmids targeting HPV 16 and 18 E6/E7 antigens (VGX-3100) and a recombinant IL-12 encoding a molecular adjuvant, delivered through electroporation with the CELLECTRA device.	Phase Ib/II trial in HPV-positive head and neck squamous cell carcinoma. Cohort 1 received vaccine pre and post surgery. Cohort 2 received vaccine after chemoradiation.Results: INO-3112 can safely general HPV-specific CD8 T-cell immunity	NCT02163057 [58]
HARE-40	Synthetic RNA-based vaccine encapsulated in RNA-lipoplex for selective update in lymphoid compartments.	Phase I/II trial of mRNA-based vaccine against the E6 and E7 oncoproteins in combination with and without anti-CD40 antibody in HPV 16-positive head and neck cancers.	NCT03418480 (active)
Whole cell based	Autologous tumor cell-based vaccines made of identified neoantigens to target the unique mutanome of each individual tumor. Advantage: personalized and able to target unique antigens Disadvantage: expensive and time consuming to manufacture		
MVX-ONCO-1	Uses irradated autologous tumor cells expressing GM-CSF combined with encapsulated cellular technology to allow continuous production of GM-CSF	Phase II trial in patients with incurable HPV-positive head and neck cancers to receive personalized cancer vaccine.	NCT02999646 (recruiting)
AlloVax	AlloVax is a personalized anti-cancer vaccine combining chaperone rich cell lysate as a source of tumor antigen prepared from patient tumors and AlloStim as an adjuvant.	Phase II trial in patients with chemotherapt refractory HPV-positive head and neck cancers to receive personalized cancer vaccine.	NCT01998542 (completed)

## 6. Conclusions

In conclusion, HPV-related cancers lend themselves to both prophylactic and therapeutic vaccines. The former is highly effective with demonstrated long-term safety. Available data strongly support the ability of approved vaccines to influence the incidence of OPSCC. The greatest challenge to HPV-preventative vaccines has been implementation, spurring universal vaccination programs in many countries. Development of therapeutic HPV vaccines in OPSCC has been challenging but several strategies are under active investigation.

## Figures and Tables

**Table 1 cancers-14-00033-t001:** Comparison of current available prophylactic HPV vaccines.

Vaccine	HPV Types Targeted	Indications	Vaccination Schedule		FDA Approval Timeline
Cervarix (2 valent)	HPV 16, 18	Females aged 9–25 years old for the prevention of cervical cancer, CIN grade 2 or worse and adenocarcinoma in situ, and CIN grade 1	0, 1, 6 months		2009: females aged 9–25 years old
Gardasil (4 valent)	HPV 6, 11, 16, 18	Females aged 9–26 years old for the prevention of cervical, vulvar, vaginal, and anal cancer, genital warts, and precancerous or dysplastic lesions	0, 2, 6 months		2006: females aged 9–26 years old 2009: males aged 9–26 years old
		Males aged 9–26 years old for the prevention of genital warts, anal cancer, and precancerous or dysplastic lesions			
Gardasil 9 (9 valent)	HPV 6, 11, 16, 18, 31, 33, 45, 52, 58	Females aged 9–45 years old for the prevention of genital warts, precancerous or dysplastic lesions and cervical, vulvar, vaginal, anal, oropharyngeal and other head and neck cancers	Age 9–14 years old	2 dose series (0 and 6–12 months) or 3 dose series (0, 2 and 6 months)	2014: females aged 9–26 years old and males aged 9–15 years old 2015: expanded for males aged 16–26 years old 2018: expanded for individuals aged 27–45 years old 2020: for prevention of certain HPV-related head and neck cancers
		Males aged 9–45 years old for the prevention of genital warts, precancerous or dysplastic lesions and anal, oropharyngeal and other head and neck cancers	Age 15–45 years old	3 dose series (0, 2 and 6 months)	

CIN, cervical intraepithelial neoplasia.

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
