# Peer review of "Vaccine Strategies for Human Papillomavirus-Associated Head and Neck Cancers"

_cancers, 2021, doi:10.3390/cancers14010033_

Round 1

Reviewer 1 Report

Zhou and colleagues garnered crucial information on HPV-associated cancers, with a particular focus on HPV-associated OPSCC, in which the number of incidences is on an alarming rise worldwide. The article provided a brief description of the epidemiology and biology of HPV-related cancers, the history and development of FDA approved HPV Prophylactic vaccines and clinical trials conducted to evaluate their safety and efficacy. This review also includes the pros and cons of different types of therapeutic vaccines available. 

According to the title, this review article focuses on HPV-associated head and neck cancer (HNC). However, under almost all sections, authors include other HPV-related cancers, especially cervical cancer. It is unclear whether the authors want to focus on HPV-related cancers or HPV-associated HNC. Under the subtitle epidemiology and biology of HPV-related cancer, the authors may consider highlighting the distinctive biology of HPV-associated HNC/OPSCC that differ from HPV-associated cervical cancer. In addition, there are other points that the authors may consider discussing/modifying, as listed below:

Line 28 - 29. Even though most papers suggest p16 as a surrogate marker for HPV-associated OPSCC, there are some controversial findings. E.g. reported by Nopmaneepaisarn et al, BMC Cancer 2019; Arsa et al., BMC Cancer 2021. Can authors discuss this or modify the tone of this statement? 

Line 55 - 58. This sentence sounds grammatically incorrect and confusing. Authors may consider clarifying the statement.

Line 63. Why did the authors mention "viral genome replication (E1 and E2/E4)"? There's E1^E4. 

Line 131. "Among women who were HPV DNA vaccine type specific positive at the time of vaccination". Authors may consider modifying or clarifying this sentence further. 

Line 168 - 169. "Additionally, ethnic minorities in the 168
United States were found to be more likely to initiate but less likely to follow through with the full HPV vaccine series". Do the authors mean initiate the uptake of the HPV vaccine? Perhaps the authors can consider discussing why some countries decided to terminate HPV vaccination programs, e.g. in Japan. 

Line 181 - 182. "The E6 and E7 proteins interact with the p53 and retinoblastoma (Rb) proteins, which are important cell cycle regulators." has been mentioned and seems redundant. 

Line 215 - 217. Please clarify this sentence: "Vaccine manufacturers have since withdrawn support for the REALISTIC trial, a phase 1 dose finding trial of Lm-based vaccine for patients with HPV 16 oropharyngeal carcinoma (NCT01598792)."

 Line 256 – 257. Please clarify this: “MHC class II-associated CD4+ helper T cells may also be activated if professional APCs phagocytose transfected somatic cells”.

Line 279 – 280. “Autologous tumor cell-based vaccines ensure that specific tumor specific antigens are preserved and used to elicit an immune response.” 2 “specific” in one sentence. Any redundancy?

Line 294 – 295. May authors clarify this: “With the success of immune checkpoint blockade a renewed interest in different vaccine platforms has emerged in combination with these agents.”?

Authors may consider summarising the different types of therapeutic vaccines, their advantages and disadvantages, effects, strategies and clinical trials findings in a table format. Perhaps, the authors may comment on the different types of therapeutic vaccines and which of them could be a more promising therapeutic option compared to others.

Minor grammatical errors are seen throughout the review article.

Original research articles should be cited, rather than citing review article.

Author Response

Zhou and colleagues garnered crucial information on HPV-associated cancers, with a particular focus on HPV-associated OPSCC, in which the number of incidences is on an alarming rise worldwide. The article provided a brief description of the epidemiology and biology of HPV-related cancers, the history and development of FDA approved HPV Prophylactic vaccines and clinical trials conducted to evaluate their safety and efficacy. This review also includes the pros and cons of different types of therapeutic vaccines available. 

According to the title, this review article focuses on HPV-associated head and neck cancer (HNC). However, under almost all sections, authors include other HPV-related cancers, especially cervical cancer. It is unclear whether the authors want to focus on HPV-related cancers or HPV-associated HNC. Under the subtitle epidemiology and biology of HPV-related cancer, the authors may consider highlighting the distinctive biology of HPV-associated HNC/OPSCC that differ from HPV-associated cervical cancer. In addition, there are other points that the authors may consider discussing/modifying, as listed below:

Line 28 - 29. Even though most papers suggest p16 as a surrogate marker for HPV-associated OPSCC, there are some controversial findings. E.g. reported by Nopmaneepaisarn et al, BMC Cancer 2019; Arsa et al., BMC Cancer 2021. Can authors discuss this or modify the tone of this statement? 

The tone of this sentence has been modified accordingly.

Line 55 - 58. This sentence sounds grammatically incorrect and confusing. Authors may consider clarifying the statement.

This sentence has been revised for clarification.

Line 63. Why did the authors mention "viral genome replication (E1 and E2/E4)"? There's E1^E4. 

This statement has been edited to “(E1, E2, and E4)”

Line 131. "Among women who were HPV DNA vaccine type specific positive at the time of vaccination". Authors may consider modifying or clarifying this sentence further. 

This sentence has been revised for clarification.

Line 168 - 169. "Additionally, ethnic minorities in the United States were found to be more likely to initiate but less likely to follow through with the full HPV vaccine series". Do the authors mean initiate the uptake of the HPV vaccine? Perhaps the authors can consider discussing why some countries decided to terminate HPV vaccination programs, e.g. in Japan. 

This sentence has been revised. This sentence describes the study of HPV vaccination among ethnic minorities in the U.S.

Line 181 - 182. "The E6 and E7 proteins interact with the p53 and retinoblastoma (Rb) proteins, which are important cell cycle regulators." has been mentioned and seems redundant. 

This sentence has been removed.

Line 215 - 217. Please clarify this sentence: "Vaccine manufacturers have since withdrawn support for the REALISTIC trial, a phase 1 dose finding trial of Lm-based vaccine for patients with HPV 16 oropharyngeal carcinoma (NCT01598792)."

This was explained in the sentence prior: “alarming safety signals have also been reported with two patients suffering from systemic listeriosis.”

 Line 256 – 257. Please clarify this: “MHC class II-associated CD4+ helper T cells may also be activated if professional APCs phagocytose transfected somatic cells”.

This sentence has been revised for clarification. An addition sentence was added for further explanation.

Line 279 – 280. “Autologous tumor cell-based vaccines ensure that specific tumor specific antigens are preserved and used to elicit an immune response.” 2 “specific” in one sentence. Any redundancy?

First “specific” was removed from sentence.

Line 294 – 295. May authors clarify this: “With the success of immune checkpoint blockade a renewed interest in different vaccine platforms has emerged in combination with these agents.”?

This sentence has been removed to avoid confusion.

Authors may consider summarising the different types of therapeutic vaccines, their advantages and disadvantages, effects, strategies and clinical trials findings in a table format. Perhaps, the authors may comment on the different types of therapeutic vaccines and which of them could be a more promising therapeutic option compared to others.

Table 1 has been added to the manuscript.

Minor grammatical errors are seen throughout the review article.

Original research articles should be cited, rather than citing review article.

Reviewer 2 Report

The manuscript is a review on vaccine strategies for HPV-associated Head & Neck cancers (HNC); anti-HPV prophylactic and therapeutic vaccines are considered. The topic is of interest (and particularly relevant in the USA), but the contents of the manuscript do not respond to the aim: 1) the informations are not centered to the cancer of interest, but describe in a general way all the HPV-associated cancers; 2) the chapter on prophylactic HPV vaccination reports data on the prevention of cervical/genital cancers, and only a brief information on the efficacy to prevent oral infection, with only one study referred, although other papers have been published; 3) the chapter on therapeutic HPV vaccines reports several trials, but they are neither appropriately described nor commented; 4) no tables or figures to summarize the findings are included.

There are also some inaccuracies:

-HPV16 etc are types (not subtypes); please, amend;

-in line 82 the period between HPV infection and OPSCC development is defined a latency period, but the latency corresponds to the time in which the virus is present at a very low copy number and is inactive

-line 121: "efficacy in preventing vaccine-related..." should be "efficacy in preventing cervical vaccine-related..."

-line 139: HPV53 is not among the types prevented by the nine-valent vaccine; please, change 53 with 52

-line 166: "Less than 2%..." is referred to a few years ago and should be substituted by more recent data

-lines 266-267: the sentence needs to be completed or modified

Author Response

The manuscript is a review on vaccine strategies for HPV-associated Head & Neck cancers (HNC); anti-HPV prophylactic and therapeutic vaccines are considered. The topic is of interest (and particularly relevant in the USA), but the contents of the manuscript do not respond to the aim: 1) the informations are not centered to the cancer of interest, but describe in a general way all the HPV-associated cancers; 2) the chapter on prophylactic HPV vaccination reports data on the prevention of cervical/genital cancers, and only a brief information on the efficacy to prevent oral infection, with only one study referred, although other papers have been published; 3) the chapter on therapeutic HPV vaccines reports several trials, but they are neither appropriately described nor commented; 4) no tables or figures to summarize the findings are included.

There are also some inaccuracies:

-HPV16 etc are types (not subtypes); please, amend;

This has been updated.

-in line 82 the period between HPV infection and OPSCC development is defined a latency period, but the latency corresponds to the time in which the virus is present at a very low copy number and is inactive

The term “latency” has been removed.

-line 121: "efficacy in preventing vaccine-related..." should be "efficacy in preventing cervical vaccine-related..."

This has been updated.

-line 139: HPV53 is not among the types prevented by the nine-valent vaccine; please, change 53 with 52

This typo has been corrected.

-line 166: "Less than 2%..." is referred to a few years ago and should be substituted by more recent data

This has been updated.

-lines 266-267: the sentence needs to be completed or modified

This has been updated.

Reviewer 3 Report

Dear Authors

Thank you for the opportunity to review the manuscript entitled « Vaccine Strategies for Human Papillomavirus (HPV) Associated Head and Neck Cancers»

Νovel approaches in HPV related cancer prevention diagnosis are one of the hottest topics in nowadays research and similar studies are needed.

My impression is that the particular well written and presented review covers all recent available aspects of HPV related cancers and should be considered for publication in your prestigious journal.

Comment

Please add a paragraph after line 136 describing any potential effect on HPV status in oropharynx after HPV vaccination based on the following article and use it as a reference too (Valasoulis G, Pouliakis A, Michail G, Kottaridi C, Spathis A, Kyrgiou M, Paraskevaidis E, Daponte A. Alterations of HPV-Related Biomarkers after Prophylactic HPV Vaccination. A Prospective Pilot Observational Study in Greek Women. Cancers (Basel). 2020 May 5;12(5):1164. doi: 10.3390/cancers12051164. PMID: 32380733; PMCID: PMC7281708)

Author Response

Comment

Please add a paragraph after line 136 describing any potential effect on HPV status in oropharynx after HPV vaccination based on the following article and use it as a reference too (Valasoulis G, Pouliakis A, Michail G, Kottaridi C, Spathis A, Kyrgiou M, Paraskevaidis E, Daponte A. Alterations of HPV-Related Biomarkers after Prophylactic HPV Vaccination. A Prospective Pilot Observational Study in Greek Women. Cancers (Basel). 2020 May 5;12(5):1164. doi: 10.3390/cancers12051164. PMID: 32380733; PMCID: PMC7281708)

We have addended the manuscript to include this study in the section “Use of HPV prophylactic vaccination around the world”

Round 2

Reviewer 2 Report

The authors have improved the manuscript, that now better respond to the declared aim. The Table clearly summarizes the trials on therapeutic vaccines under investigation in patients with oropharyngeal/head & neck cancers.

MINOR COMMENTS:

Lines 193-194: please, check the sentence; the words "vectors" and "vaccines" should probably be interchanged.

Table 1: I would suggest to rephrase the title as "Therapeutic HPV vaccines under development in trials for oropharyngeal/head and neck cancers" (eight minor spelling mistakes are present in the table).
